# Simple Fabrication of Photodetectors Based on MoS_2_ Nanoflakes and Ag Nanoparticles

**DOI:** 10.3390/s22134695

**Published:** 2022-06-22

**Authors:** Peng Xiao, Ju-Hyung Kim, Soonmin Seo

**Affiliations:** 1College of BioNano Technology, Gachon University, Seongnam 13120, Korea; zhongpengxiao@gmail.com; 2Department of Chemical Engineering, Ajou University, Suwon 16499, Korea; 3Department of Energy Systems Research, Ajou University, Suwon 16499, Korea

**Keywords:** transition-metal dichalcogenides (TMDs), MoS_2_ nanoflakes, MoS_2_ photodetectors, visible light sensor

## Abstract

Low-dimensional transition-metal dichalcogenides (TMDs) have recently emerged as promising materials for electronics and optoelectronics. In particular, photodetectors based on mono- and multilayered molybdenum disulfide (MoS_2_) have received much attention owing to their outstanding properties, such as high sensitivity and responsivity. In this study, photodetectors based on dispersed MoS_2_ nanoflakes (NFs) are demonstrated. MoS_2_ NFs interact with Ag nanoparticles (NPs) via low-temperature annealing, which plays a crucial role in determining device characteristics such as good sensitivity and short response time. The fabricated devices exhibited a rapid response and recovery, good photo-responsivity, and a high on-to-off photocurrent ratio under visible light illumination with an intensity lower than 0.5 mW/cm^2^.

## 1. Introduction

Low-dimensional transition-metal dichalcogenides (TMDs) have received considerable attention as promising materials for high-performance semiconductors in several fields such as electronics, optoelectronics, and energy conversion and storage [1,2,3,4,5,6,7,8]. In particular, extensive research has been conducted to explore molybdenum disulfide (MoS_2_), which has a direct bandgap structure contributing to its excellent electrical and optical properties [9,10]. The fundamental properties of MoS_2_, such as quantum confinement effects and the functions of chalcogen vacancies and photoexcited carriers, have been extensively studied for their practical and potential applications [11,12,13,14,15]. Several electronic and optoelectronic devices, including field-effect transistors, charge-trap memories, and photodetectors based on mono- and multilayered MoS_2_, have been realized [16,17,18,19]. Interestingly, thin layers of MoS_2_ with structural integrity exhibit high light-absorption coefficients, enabling sensitive photon detection via photoexcited carriers. These thin layers are conventionally fabricated using chemical vapor deposition (CVD) growth methods [20,21,22]. Recently, solution-phase exfoliation and stabilization of TMDs have been proposed as alternative methods that facilitate facile and rapid fabrication [23,24,25,26]. These methods enable synthesis of MoS_2_ with small dimensions, such as nanoflakes (NFs), in a size-controlled manner, demonstrating their significant advantages such as low-temperature and solution-based processing. However, in the context of photodetecting, connectivity between MoS_2_ nanostructures with high light-absorption coefficients is required during film formation to provide conducting pathways for the collecting photoexcited carriers generated in MoS_2_ through electrodes. Thus, considerable research is underway to achieve high electrical conductivity and environmental stability in nanostructured MoS_2_ films. For example, band-selective photodetection has been successfully demonstrated using TMD nanosheets exfoliated with amine-terminated polymers in solution [27].

In this study, thin photodetectors based on dispersed MoS_2_ NFs are presented, and the optoelectronic properties for high absorption coefficients of photons are discussed in relation to experimental and theoretical studies. Initially, MoS_2_ NFs are synthesized in the solution phase with planar dimensions of ~20 nm for thin film formation via the drop-coating method, using Ag nanoparticles (NPs) formed from Ag nanofilms to ensure electrical connectivity between the MoS_2_ NFs. The Ag contacts to bulk MoS_2_ layers are known to become ohmic, and the resistivity of the MoS_2_ layers decreases after annealing at 400–600 °C [28]. Furthermore, Ag NPs and islands can enhance light absorption in photoresponsive devices via plasmonic [29,30]. Thus, the photoexcited carriers in MoS_2_ can be efficiently collected even under low-intensity illumination (<0.5 mW/cm^2^) with rapid response and recovery times. Photodetectors based on MoS_2_ NFs and Ag NPs have exhibited good photodetection abilities under low-intensity illumination and ambient conditions. This study provides a novel and simple route to achieve good performance thin photodetectors with desirable photodetection abilities for practical applications, particularly under low-intensity illumination conditions. These results demonstrate great potential for the development of advanced light-sensing systems, and other TMD-based applications such as transistors, memories, and energy conversion and storage devices.

## 2. Materials and Methods

### 2.1. Preparation of MoS_2_ NFs Solution

MoS_2_ NFs were prepared using a simple liquid exfoliation technique; 500 mg of MoS_2_ micro-crystals (Sigma Aldrich, St. Louis, MO, USA) were dispersed in 250 mL of a deionized (DI) water:ethyl alcohol (1:1) solution. The mixture was sonicated using a probe sonic tip (19 mm in diameter) at a power of 500 W using 10 s ON and 10 s OFF pulses for 10 h. DI water:ethyl alcohol solution (1:1, 50 mL) was added into the beaker every 2 h to maintain the volume of the MoS_2_ solution (250 mL) because the solution easily evaporated during ultrasonication owing to the high temperature. Additionally, the continuous feeding of the solution prevented a steep increase in temperature during the ultrasonication process. Subsequently, the solution was diluted to 1000 mL and centrifuged for 5 min at 5000 rpm to separate and remove the unexfoliated MoS_2_. The supernatant solution was further centrifuged at 8000 rpm for 30 min to remove large MoS_2_ particles. Approximately 90% of the supernatant solution was filtered under vacuum onto a membrane paper with a pore size of 100 nm. The filtered solution was dried in an oven to concentrate the MoS_2_ NF solution, until the volume of the solution reached 500 mL.

### 2.2. Photodetector Fabrication

Photodetectors were fabricated based on the schemes shown in Figure 1a. For example, the photodetector (20 nm Ag, 400 °C, vacuum, 24 h) exhibited high performance. In the typical procedure for the bottom-mode photodetector, a rectangular SiO_2_ (300 nm)/Si substrate with dimensions of 1 × 1 cm^2^ was cleaned using oxygen plasma equipment. The substrate was drop coated with the MoS_2_ NF solution at a solution/substrate ratio of ~0.3 mL/cm^2^. Subsequently, the solution was dried on the substrate in an oven at 80 °C for 30 min. The top Ag thin film was evaporated onto the sample under vacuum (<2 × 10^−6^ Torr) using a thermal evaporation system. The prepared sample was then annealed at 400 °C for 24 h in a muffle furnace at 1 atm, using a heating rate of 50 °C/min. After cooling to room temperature, the Au electrodes (50 nm) were evaporated onto the thin annealed film under vacuum using a patterned shadow mask. For the middle-mode photodetector, a 10 nm Ag layer was deposited on the SiO_2_ substrate and annealed at 400 °C for 24 h at atmospheric pressure. The MoS_2_ solution was then drop coated and dried in an oven at 80 °C for 30 min. A second 10 nm thick Ag film was deposited on the dried sample and annealed under the same conditions. Subsequently, the Au electrodes (50 nm) were evaporated onto the thin annealed film, as described above. An optical image of the fabricated device is shown in Figure 1c.

### 2.3. Process and Measurements

The following equipment was used: ultrasonicactor (SONICS-VCX500, SONICS, Newtown, CT, USA), centrifuge (UNIVERSAL 320R, Hettich, Beverly, MA, USA), field emission scanning electron microscope (FESEM, JSM-7500F, JEOL, Tokyo, Japan), vacuum muffle furnace (Neytech Qex, DEGUSSA-NEY DENTAL, INC., Bloomfield, CT, USA), light source (FOK-100W, Fiber Optic Korea, Cheonan, Korea), photodetectivity measurement system (4200-SCS, Keithley, Beaverton, OR, USA), thermal evaporation system, and solar simulator for the on/off test (XES-301S, SAN-EI ELECTRIC CO., Ltd., Osaka, Japan).

## 3. Results and Discussions

As the solvent of the MoS_2_ NFs solution was slowly evaporated, the MoS_2_ NFs were aggregated in a thin film form on the substrate. The surface morphology of the MoS_2_ film after solvent evaporation was examined as shown in Figure 1d. Scanning electron microscope (SEM) and atomic force microscopy (AFM) images clearly revealed that the MoS_2_ NFs were agglomerated, forming continuous domains. The thickness of the MoS_2_ film was measured to be ~20 nm, indicating the multilayered MoS_2_ domains. Size distribution of the MoS_2_ NFs in the solution is also shown in Figure 1e. Diameters of the NFs were mostly smaller than 300 nm, and it was also found that the most probable diameter of the NFs was 150 nm. In addition, Raman spectroscopy was employed to clarify the existence of the MoS_2_ NFs in the film without significant changes in chemical composition. Two prominent peaks were observed around 381 cm^−1^ and 409 cm^−1^, as shown in Figure 1f. These peaks can be assigned to the multilayered MoS_2_ in consideration of the peak shifts, which are typically observed in the multilayered MoS_2_. It is worth noting that the center of the thin film was not completely covered with the MoS_2_ NFs, due to the coffee-ring effect that occurred in the drop-casting process, and thus the substrate was partially exposed to air.

Figure 2 shows the SEM micrographs of the annealed Ag nanofilm on MoS_2_ layers at various temperatures (without annealing and annealed at 200, 400, 450, 500, and 550 °C). All samples were annealed for 24 h at atmospheric pressure. The surface of the Ag nanofilm (20 nm thick) on the MoS_2_ layer before the annealing process was flat, and some dark spots appeared on the surface, as shown in Figure 2a. The dark spots are slightly dented and considered to be an uncovered region of the MoS_2_ film. After annealing, the Ag film on MoS_2_ was dewetted and formed Ag islands on the surfaces, as shown in Figure 2b–f. Exfoliated MoS_2_ nanosheets are known to be electrically conductive [31]; however, the electrical conductivity of MoS_2_ films after the drop-coating process was low because there are many inter-nanosheet junctions between the MoS_2_ nanosheets, which complicates electron conduction between them at low voltages [32]. By annealing the Ag film, Ag diffused into the MoS_2_ layer, increasing the MoS_2_ conductivity. Furthermore, the contact between the metal and the MoS_2_ layer is known to become ohmic after diffusion, reducing the resistivity [33]. Thus, annealing of the Ag film improved the electrical contact between the MoS_2_ layers. In particular, the device, which was solely based on MoS_2_ without Ag, did not show any photo response. As mentioned above, this result is possibly originating from the inter-nanosheet junctions reducing the electrical conductivity. The MoS_2_ NFs were also concentrated at the edge of the thin film due to the coffee-ring effect in the drop-casting process, causing domain discontinuities near the center. Such disconnection between MoS_2_ disrupts the electron transportation and the photo response. However, when the Ag nanofilm was formed and annealed on the sample, the MoS_2_ domains were well connected to each other through the Ag islands consisting of the Ag NPs. The radii of the Ag islands were measured to be in the range of 10–600 nm, and the particle size increased with the annealing temperature (See Figure 2). It was also observed that small Ag NPs (<10 nm) were attached to the MoS_2_ surface, leading to connection between Ag and MoS_2_ after annealing. These results are consistent with the previous studies reporting that Ag can be easily attached to the MoS_2_ NFs. It has been reported that the Ag NPs can be attached to the MoS_2_ surface via the formation of the Ag ions in the solution [34], and the MoS_2_ layer can play a role as a photocatalyst with the Ag NPs [15]. Notably, the surface plasmon resonance effect of metal NPs can increase visible light absorption [35,36]. Surface plasmons can be localized by Ag NPs, and the excitation of localized surface plasmon resonance can occur. This leads to an enhanced electric field, which increases the photocurrent of the MoS_2_ layer.

Thin photodetectors based on MoS_2_ NFs and Ag NPs were fabricated as shown in Figure 1a. The two types of device structures, bottom-mode and middle-mode structures, are also described in Figure 1b. Prior to measuring performance of the Ag-coated MoS_2_ devices, a device with only the MoS_2_ film on the SiO_2_/Si substrate was fabricated that was annealed at various temperatures from 200 to 1000 °C in a muffle furnace. However, no photo response was observed in the device. Subsequently, two bottom-mode devices were fabricated with 20 and 110 nm thick Ag films. The Ag nanofilm on the MoS_2_ layer was annealed at 400 °C. The characteristic curves of the devices are shown in Figure 3a,b. Additionally, a device with a 200 nm Ag film was fabricated. However, the results for this device are not presented because the Ag layer was too thick, and Ag did not form islands after annealing. Thus, the device with a 200 nm Ag film short circuited and exhibited a maximum current of 1 × 10^−2^ A at all voltages.

Subsequently, the channel width of the Au electrodes was varied from 90 to 1000 μm, as shown in Figure 3a,b. The characteristic current vs. voltage curves of the annealed device with the 20 nm Ag film are shown in Figure 3a, and those of the annealed device with the 110 nm Ag film are shown in Figure 3b. The devices were exposed under illumination using a visible light source (OSRAM, Munich, Germany, 64637) with a power density of 14.1 mW/cm^2^. All the devices were photoresponsive, and the devices were not strongly dependent on the channel width. As shown in Figure 3c, both devices with 20 and 110 nm Ag films show dark currents below 1 × 10^−6^ A and on-currents at 10 V over 1 × 10^−4^ A. Moreover, the on/off current ratios of the devices with 20 and 110 nm Ag films are shown in Figure 3c. The fabrication conditions (20 nm Ag film thickness and 500 μm channel width) showed the best performance and were adopted for further experiments. As shown in Figure 3d, the on/off ratio was affected the performance of the devices based on the annealing temperature conditions. The highest on-current at 10 V was observed for the device annealed at 450 °C. However, the off-current was also relatively higher than those of the devices annealed at 300 and 400 °C. Among them, the device annealed at 400 °C, which had the highest on/off ratio (1.66 × 10^3^), was selected for the fabrication process for better results in this study.

To evaluate the photodetection properties of the device annealed at 400 °C, the output characteristics and photoresponsivity were measured under visible light illumination with various intensities, as shown in Figure 4a. For the photoresponsive measurements, two sharp probes were brought into contact with the Au electrodes of the device. As mentioned previously, the device with 20 nm of Ag film and annealed at 400 °C showed the highest performance among the fabricated devices. Its photoresponsivity was 4.37 × 10^1^ AW^−1^ under low-intensity illumination (~0.5 mW/cm^2^) and decreased to 1.53 × 10^1^ AW^−1^ at a light intensity of 14.1 mW/cm^2^. The photocurrent gradually increased as the light intensity increased, and the photoresponsivity remained constant at ~1.5 × 10^1^ AW^−1^. It is noteworthy that the photocurrent increased to ~11 mA at a sample bias voltage of 10 V under high-intensity illumination (14.1 mW/cm^2^). The time-resolved photo response of the device is shown in Figure 4b. The photocurrent of the device rapidly changed from 1.7 × 10^−5^ to ~2.5 × 10^−3^ A in response to the on/off switching of light illumination (126 mW/cm^2^) at a constant sample bias voltage of 10 V. The photo response and recovery times were consistently measured as ~324 and ~262 ms, respectively, because the on- and off-currents instantaneously returned to their initial levels without any losses. It is noteworthy that the shutter speed of the light source (a few milliseconds) was not compensated, and thus, the real response times were probably shorter than the measured values. The reversibility of the photoresponsive device is shown in the inset in Figure 4b. As the light illumination switches on and off, the initial off-current and on-current values are repeatedly obtained.

Based on the results, it was difficult to increase the on/off ratio above 2.0 × 10^3^ using the MoS_2_-Ag photoresponsive materials formed under the investigated conditions. Although the on-current of the device with a 10 nm thick Ag film was lower than that of the other fabricated devices with thicker Ag films, it exhibited a higher on/off ratio at 10 V compared with the other devices. Based on this result, another device configuration was evaluated to increase the on/off ratio. The middle-mode structure of the light-sensing device was designed, as shown in Figure 1b. To increase the light absorption of the device while maintaining a high on/off ratio, an Ag nanofilm was deposited and annealed before the MoS_2_ drop-coating process. This structure is called a ‘middle-mode’ structure, and the structure of previously discussed is called a ‘bottom-mode’ structure, as shown in Figure 1b. The designation of the structures was determined by the position of the MoS_2_ layer on the Ag films. In this study, the thickness of the Ag film below the MoS_2_ layer was 10 nm, and the same Ag film thickness was used for the device with the middle-mode structure. All Ag nanofilms of the devices were annealed immediately after deposition, as described previously. Particularly, the off-current of the device with 10 nm of Ag film annealed at 400 °C had a low value to be distinguished, compared with the devices with thicker Ag films. Thus, the middle-mode structure is suggested for increasing the on/off ratio of the photoresponsive device.

The characteristic curves of the two different devices annealed at 400 °C are shown in Figure 5a. The off-current of the bottom-mode device (Ag (20 nm)/MoS_2_/substrate) at a sample bias voltage of 10 V was relatively higher than the middle-mode device (Ag (10 nm)/MoS_2_/Ag (10 nm)/substrate), which was annealed at the same temperature. The device with the middle-mode structure showed a lower off-current, which resulted in a high on/off current ratio at 10 V. As shown in Figure 5b, the on-current of the middle-mode device under illumination with a visible light intensity of 14.1 mW/cm^2^ and a sample bias voltage of 10 V was 6.88 × 10^−5^ A, and the off-current under the same conditions was 1.25 × 10^−9^ A. Thus, the middle-mode device annealed at 400 °C exhibited low on- and off-current values, and the on/off current ratio was 5.61 × 10^4^ at 10 V, which is higher than that of the bottom-mode device with a 20 nm Ag film annealed at the same temperature. As shown in Figure 5b, the photocurrent of the middle-mode device gradually increased as the light intensity increased from 0 to 14.1 mW/cm^2^ at a sample bias voltage of 10 V. The photoresponsivity was 1.17 AW^−1^ under low-intensity illumination (~1.8 mW/cm^2^) and decreased to 9.75 × 10^−1^ AW^−1^ at a light intensity of 14.1 mW/cm^2^.

Another device was fabricated under different annealing conditions to reduce the annealing temperature. The characteristic curves of the two different devices annealed at 300 °C are shown in Figure 5c. The off-current of the middle-mode device at a sample bias voltage of 10 V was lower than that of the bottom-mode device with a 20 nm Ag film annealed at 300 °C. The middle-mode device exhibited a higher on/off current ratio at a sample bias voltage of 10 V, compared with the bottom-mode device. As shown in Figure 5d, the on-current at a sample bias voltage of 10 V under illumination with a visible light intensity of 14.1 mW/cm^2^ was 1.74 × 10^−4^ A, and the off-current under the same conditions was 1.22 × 10^−8^ A. The middle-mode device annealed at 300 °C showed low on- and off-current values, and the on/off current ratio was 1.42 × 10^4^, which is 82.7 times higher than that of the bottom-mode device with a 20 nm Ag film annealed at 300 °C. The photocurrent of the middle-mode device at a sample bias voltage of 10 V gradually increased as the light intensity increased from 0 to 14.1 mW/cm^2^, as shown in Figure 5d. The photoresponsivity was 8.10 × 10^−2^ AW^−1^ under low-intensity illumination (~1.8 mW/cm^2^) and increased to 2.47 AW^−1^ at a light intensity of 14.1 mW/cm^2^. Moreover, switching test was performed using the middle-mode device, of which the MoS_2_ film was sandwiched between the two Ag films (i.e., top and bottom). Each Ag film was 10 nm thick, and annealed at 300 °C. The device was repeatedly exposed to the visible light with an intensity of 14.1 mW/cm^2^ for 60 s. The current rapidly increased and decreased in response to the light as shown in Figure 5e. The device responded 43 cycles during 60 s, and the on-current was consistently maintained at ~1.6 × 10^−4^ A on average.

Based on these results, the on/off current of the device can be increased by fabricating a middle-mode structure and decreasing the annealing temperature to enhance the performance. Although the performance was not very high, the device annealed at 300 °C exhibited sufficient performance for application as a photoresponsive device. As reported in a previous study, Ag NPs located below the MoS_2_ layer enhanced the light absorption in the photoresponsive system [35]. In this study, a dewetted Ag film formed NPs on a MoS_2_ layer, which enhanced light absorption in the system. Thus, the dark current value was not significantly changed, and the on-current greatly increased owing to the enhanced light absorption by the Ag NPs below the active layer, resulting in an increase in the on/off ratio of the photoresponsive device.

## 4. Conclusions

Photodetectors were developed based on MoS_2_ NFs as the conversion center to transfer energy from photons to electrons in the thin film. In this study, after annealing, Ag diffused into the MoS_2_ layer, which decreased the resistance of the inter-nanosheet junctions between the MoS_2_ layers. Ag islands connected discontinuous MoS_2_ NFs each other. This increased the conductivity of the MoS_2_ layer. Moreover, the newly formed Ag islands on the MoS_2_ layer enhanced the absorption efficiency of light because the surface plasmon resonance effect of metal NPs increases visible light absorption. Two different device modes were fabricated for the photodetector. First, bottom-mode devices were fabricated to determine the optimal fabrication conditions based on performance. The bottom-mode device with a 20 nm Ag film annealed at 400 °C showed the highest performance, with a photoresponsivity of 4.37 × 10^1^ AW^−1^ under low-intensity illumination (~0.5 mW/cm^2^) at a sample bias voltage of 10 V, and it exhibited the highest on/off ratio (1.66 × 10^3^). Second, a middle-mode device was fabricated to increase the on/off ratio. The middle-mode device annealed at 400 °C exhibited lower off-current, which caused a high on/off current ratio of 5.61 × 10^4^ at a sample bias voltage of 10 V. Therefore, the on/off ratio increased by over an order and was 82.7 times higher than that of the bottom-mode device annealed at 300 °C. In conclusion, MoS_2_ NFs play a major role in transferring newly generated electrons to the Ag film under illumination. Therefore, the developed methodology is proposed as an effective way to capture energy from conversion centers, such as TMD NFs using nano-thick metal films. This is a crucial concept for the utilization of various NFs and TMDs in optoelectronic applications. Thus, these results are expected to contribute to the advancement of high-performance photoresponsive systems for light-sensing applications.

## Figures and Tables

**Figure 1 sensors-22-04695-f001:**
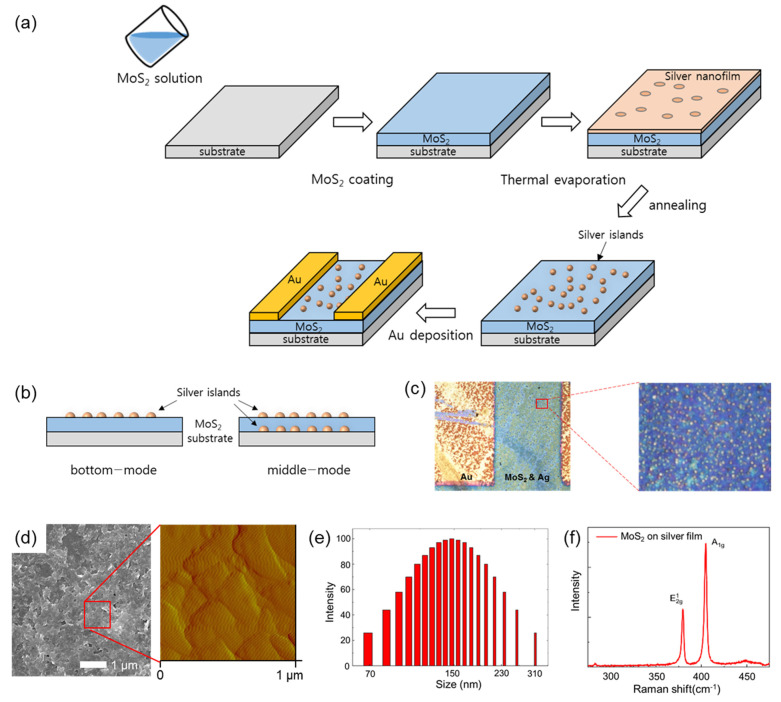
(**a**) Illustration of the fabrication of photodetecting devices. (**b**) Schematic of the bottom- and middle-mode structures. (**c**) Optical image of the fabricated device and magnified image of Ag islands on the MoS_2_ layer. (**d**) SEM image of MoS_2_ film and magnified AFM image of boxed area (**e**) Size distribution of the MoS_2_ NFs. (**f**) Raman spectra of MoS_2_ film on silver film.

**Figure 2 sensors-22-04695-f002:**
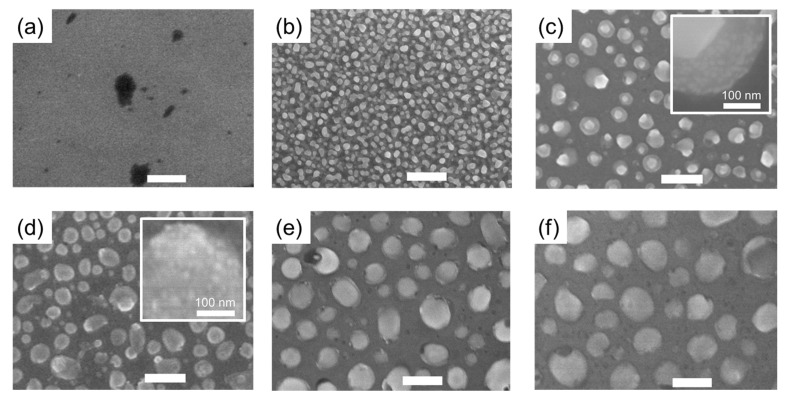
FESEM micrographs of annealed Ag nanofilms on the MoS_2_ layers at various temperatures. The thickness of the deposited Ag film was 20 nm. Samples were annealed at (**a**) room temperature (without annealing), (**b**) 200 °C, (**c**) 400 °C, (**d**) 450 °C, (**e**) 500 °C, and (**f**) 550 °C. The solid white bars represent the 1 μm length scale and the sold white bars in the inset of the (**c**,**d**) represent the 100 nm.

**Figure 3 sensors-22-04695-f003:**
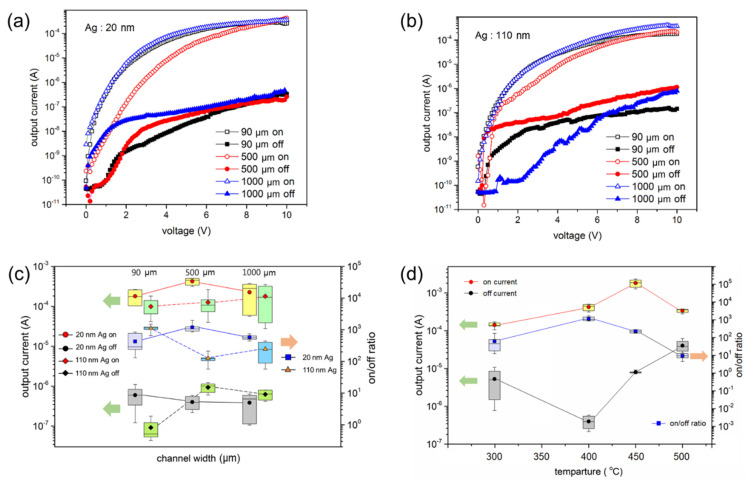
Output characteristics (output current vs. sample bias voltage) of the bottom−mode photodetecting devices under various fabricating conditions: (**a**) 20 nm and (**b**) 110 nm Ag films annealed at 400 °C with various electrode channel widths (90, 500, and 1000 μm). Output currents were measured under visible light illumination with an intensity of 14.1 mW/cm^2^ (on) and 0.0 mW/cm^2^ (off). (**c**) Output currents and on/off ratios of bottom−mode devices in (**a**,**b**) at a bias voltage of 10 V. (**d**) Output currents and on/off ratios at a bias voltage of 10 V of the bottom−mode devices with a 500 µm electrode channel width and 20 nm Ag film thickness annealed at various temperatures (300, 400, 450, and 500 °C).

**Figure 4 sensors-22-04695-f004:**
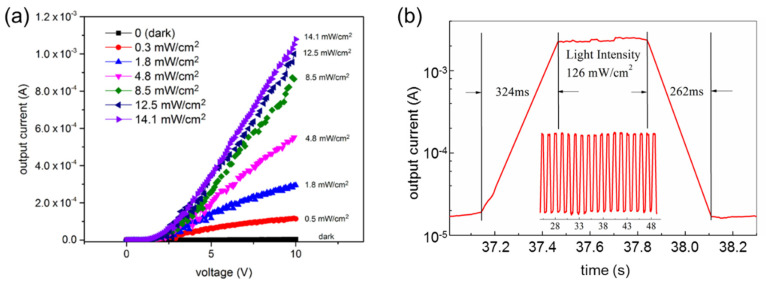
(**a**) Output characteristics of the bottom−mode photodetecting device with 20 nm of Ag film annealed at 400 °C with a 500 μm electrode channel width. Output currents were measured under visible light illumination with various intensities of 0.0 (dark), 0.5, 1.8, 4.8, 8.5, 12.5, and 14.1 mW/cm^2^. (**b**) Real-time characteristics of the output current measured in response to a light intensity of 14.1 mW/cm^2^ at a constant sample bias voltage of 10 V. The data clearly show the reversible photoresponsivity of the device.

**Figure 5 sensors-22-04695-f005:**
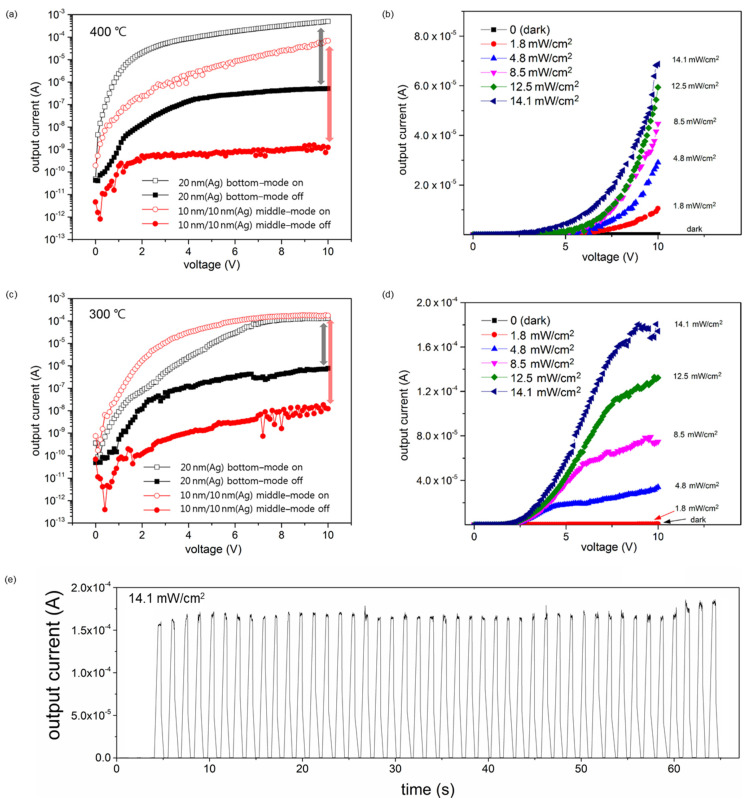
(**a**) Output characteristics of the bottom-mode device with 20 nm (top) of Ag film and middle-mode device with 10 nm (top)/10 nm (bottom) of Ag film, both annealed at 400 °C. (**b**) Output characteristics of the middle-mode device with 10 nm (top)/10 nm (bottom) of Ag film annealed at 400 °C under visible light illumination with various intensities. (**c**) Output characteristics of the bottom-mode device with 20 nm (top) of Ag film and middle-mode device with 10 nm (top)/10 nm (bottom) of Ag film, both annealed at 300 °C. (**d**) Output characteristics of the middle-mode device with 10 nm (top)/10 nm (bottom) of Ag film annealed at 300 °C under visible light illumination with various intensities. The on and off-currents of (**a**,**c**) were measured under visible light illumination with an intensity of 14.1 mW/cm^2^. The output currents of (**b**,**d**) were measured under visible light illumination with various intensities of 0.0 (dark), 1.8, 4.8, 8.5, 12.5, and 14.1 mW/cm^2^. (**e**) Switching test curve of the middle-mode device with 10 nm (top)/10 nm (bottom) of Ag film annealed at 300 °C. The device responded 43 cycles for 60 s under visible light illumination with an intensity of 14.1 mW/cm^2^.

## Data Availability

Not applicable.

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
