# Peer review of "Simple Fabrication of Photodetectors Based on MoS2 Nanoflakes and Ag Nanoparticles"

_sensors, 2022, doi:10.3390/s22134695_

Round 1

Reviewer 1 Report

The authors studies the photoresponse behaviours of devices with MoS2 nanoparticles/Ag film as the photoactive channel. With Ag film diffused into MoS2 nanoparticles, the absorption efficiency of light is enhanced, so devices show high performance under visible light illumination.

  1. The main mechanism of the photo-induced current in the fabricated devices is not clean. There are no evidences supported for the conclusion stating that “MoS2 NP play a major role in transferring newly generated electrons to the Ag film under illumination”. In fact, due to some observations, including channel-width independent current and reduced photoresponsivity with increased light intensity, the photocurrent in the device may come from modulation of the Schottky barriers at the interfaces between the Au electrodes and thin film rather than from the conducting channel. Local irradiation, focusing on different positions in the device, using different-wavelength light (compared to 1.2-eV bandgap of MoS2), may help to help to explain the underline mechanism. 
  2. No information to clarify existence of “MoS2 nanoparticles” in the fabricated devices (perhaps the authors need to provide Raman/EDX and SEM/AFM data of their MoS2 thin film coated on the substrate).
  3. The term “Ag nanofilms” is a misleading concept in this work, since the photo-detector devices work only after annealing the evaporated Ag thin film to form the clusters or small particles.
  4. There have been numerous researches on photodetectors using MoS2-based hybrid structures. The authors probably need to provide a comparison of their device performance to claim that they achieve “high-performance” photodetector.
  5. What is the wavelength of the visible light?
  6. How many devices were measured for each type of devices regarding to the device parameters? If only one device for each type, then how do the authors ensure the difference on the performance is not from the devices 
  7. The manuscript listed the performances of many devices with different fabrication parameters. It will be better and clearer if the authors can make a table to present all the key metrics of photodetection of different devices.
  8. The authors need to work more on the manuscript for the consistency and clarity:
  • Page 6, line 199, the authors mentioned ‘As shown in Fig.2, a device with a 10nm-thick Ag film on the MoS2 layer was fabricated.’, but there is no 10nm-thick Ag film devices shown in Fig.2. 
  • The title fonts of axises of all the figures need to be consistent. 
  • The results all over the manuscript are not very clearly presented. 
  •  

Author Response

Dear reviewer,

This communication is in reference to the manuscript (sensors-1757019). We changed title from “High-performance photodetectors based on MoS2 nanoparticles and Ag nanofilms” to “Simple fabrication of photodetectors based on MoS2 nanoparticles and Ag nanoparticles”. We’d like to express our heartfelt gratitude to you and the referees for kindly considering our manuscript. We thank the reviewer for his/her great efforts to review the manuscript and giving the positive evaluation on our work. We also thank you for providing us an opportunity to respond to the referees’ concerns.

We believe that we have addressed all the reviewer’s comments and technical corrections. The changes are highlighted in red in the revised manuscript. We hope that the revised manuscript resolves all the concerns raised by the reviewers and meets the quality of expected for a Sensors article.

I look forward to your final decision. Thank you very much. Please contact me if you have any questions.

Sincerely yours

Soonmin Seo

Department of Bionano Technology, Gachon University

soonmseo@gachon.ac.kr

Reviewer 2 Report

Comments:

In this manuscript, the authors have prepared the MoS2 nanoparticles and Ag nanofilms and studied their photoresponse properties. The work is not well organized and studied. Therefore, it cannot be accepted for publication.

1.     There is lack of novelty, several articles published with this material with enhanced performance and this experimental analyses are straightforward, which are not systematically studied and optimized.

2.     What is the size of MoS2 NPs? Do other solutions with different concentrations have better properties? Also, authors need to emphasis the role and detailed structure of MoS2, no data reported without Ag.

3.     There is no supporting data such as TEM, EDX, XRD and Raman to confirm this material and structure, lack of materials characterizations.

4.     Also, there is no detailed measurements on photodetection properties Current vs time, detectivity, responsivity curve and repeatability test etc.

5.     More importantly, authors did not provide clear information on the enhancement mechanism to support the result such as bandgap structure or Ag-MoS2 junction schematic representation.

6.       A good photodetector exhibits the better switch ratio and photoresponsivity at less bias (1V or 5V). Authors used 10V bias, which is not obvious in real-time photodetector.

7.     Author may add the visible light wavelength. There is no comparison with other study to emphasis the device performance. 

Due to above concerns this work is not suitable for Sensors Journal. I may reconsider if authors address the above issues.

Author Response

(The authors gave the same response as above.)

Reviewer 3 Report

(1) what is the thickness of MoS2 NPs film? please give the optical absorption spectrum of MoS2 NPs film.

(2) please clarify and demonstrate why the middle-mode devices can increase the light absorption and on-off ratio? 

(3) how many devices have been measured in this paper? Please give the statistics values.

Author Response

(The authors gave the same response as above.)

Author Response

(The authors gave the same response as above.)

Round 2

Reviewer 1 Report

The study is sound and the language is clear after the revision. I approve of its publication. 

Author Response

Response to reviewer #1

Reviewer #1: The study is sound and the language is clear after the revision. I approve of its publication. 

The comments are listed below:

Thank you for your helpful advices, we were able to improve the quality of the manuscript.

Reviewer 2 Report

Current manuscript has been improved significantly.

Author Response

Response to reviewer #2

Reviewer #2: Current manuscript has been improved significantly.

The comments are listed below:

Thank you for your helpful advices, and we were able to improve the quality of the manuscript.

Reviewer 3 Report

(1) Fig. 3 (c) and (d) in the revised manuscript should be presented in a boxplot form.

(2) the authors say" At least, three or more of the same device were made and measured twice or more". I think at least ten or more devices should be fabricated and measured.

Author Response

Response to reviewer #3

  1. (1) Fig. 3 (c) and (d) in the revised manuscript should be presented in a boxplot form.

(2) the authors say" At least, three or more of the same device were made and measured twice or more". I think at least ten or more devices should be fabricated and measured.

→ (1) As the reviewer suggested, Fig. 3(c) and (d) in the revised manuscript are presented in a boxplot form as shown in the following figure. (2) We agree with what the reviewer said, however, we are sorry that we could not respond to the correction due to insufficient time. We will refer to the reviewer's appreciated advice in future research. 

Reviewer 4 Report

The authors answered all the questions, the manuscript improved and now it is clear. In my opinion, I considered the paper accepted. However, I have a personal suggestion and one minor revision.

-About the photocurrent mechanism in TMDs material I suggest the author read about exciton annihilation in 2D material. Indeed the strong binding energy between electron and hole results in an unlucky dissociation to form a photocurrent. So the exciton population itself contributes in part to the dissociation of e-h via annihilation.

-minor: although MoS2 is bulky and not ultrathin, I do not agree with the terminology  ''nanoparticle''. Nanoparticles are not Van der Waals crystals. I would rather prefer ''multilayer MoS2 flakes''.   

Author Response

Response to reviewer #4

The authors answered all the questions, the manuscript improved and now it is clear. In my opinion, I considered the paper accepted. However, I have a personal suggestion and one minor revision.

Comments:

  1. About the photocurrent mechanism in TMDs material I suggest the author read about exciton annihilation in 2D material. Indeed the strong binding energy between electron and hole results in an unlucky dissociation to form a photocurrent. So the exciton population itself contributes in part to the dissociation of e-h via annihilation.

→ Thank the reviewer for the helpful suggestion. We will read about exciton annihilation in 2D material and it will help explain the photocurrent mechanism in TMDs material.

  1. minor: although MoS2 is bulky and not ultrathin, I do not agree with the terminology ''nanoparticle''. Nanoparticles are not Van der Waals crystals. I would rather prefer ''multilayer MoS2 flakes''.   

→ Thank the reviewer for the helpful comments. We agree with the reviewer. Throughout the manuscript, ‘MoS2 nanoparticles’ were corrected to ‘MoS2 nanoflakes’.
